# Diagnostic yield, complications, pathology and anatomical features in CT-guided percutaneous needle biopsy of mediastinal tumours

**Ingegjerd Kristina Skretting[1], Espen Asak Ruud[1,2], Haseem Ashraf [1,2,3] ***

**1** University of Oslo, Oslo, Norway, **2** Department of imaging, Akershus University Hospital, Oslo, Norway, **3** Department of Pulmonary Medicine, Gentofte University Hospital, Hellerup, Denmark

\* haseem.ashraf@gmail.com

## Abstract

### Objectives

This study presents the experiences of percutaneous CT-guided needle biopsy at a university hospital in Norway.

### Methods

A retrospective examination of all mediastinal biopsy procedures between April 2015 and August 2019 was performed at Akershus University Hospital in Norway. We registered patient and procedure characteristics, along with lesion pathology and characteristics including localization according to anatomical and Felson mediastinal compartments.

### Results

The study included 48 procedures, conducted in 45 patients (29 men and 16 women) with a mean age of 60,5 years. Pneumothorax occurred in 12 procedures (60% of the transpulmonary procedures) and pneumomediastinum in 18 procedures (38%). Pneumothorax was only seen in procedures with transpulmonal access. Four of the pneumothorax cases required pleural drainage. Diagnostic yield was 96%. We found significant (p = 0,006), moderate to high association between anatomical compartment localization and histopathological diagnosis (Cramér's V = 0,49) for tumours selected for CT-guided percutaneous biopsy. Felson's compartment division on the other hand, did not show any significant associations.

### Conclusion

We found CT-guided percutaneous needle biopsy of mediastinal tumours to be an effective and safe procedure with a diagnostic yield of 96%. The main complications were pneumothorax and pnumomediastinum, with a relatively low chest drainage rate. Anatomical mediastinum compartment showed a significant, moderate to high association with the histopathological diagnosis for tumours selected for percutaneous CT-guided biopsies, where most malignancies were seen in the anterior compartment.

**Data Availability Statement:** All relevant data are within the paper.

**Funding:** The authors received no specific funding for this work.

**Competing interests:** The authors have declared that no competing interests exist.

## Introduction

Mediastinal tumours are rare and represent only 3% of tumours in the chest [1]. In patients with lesions in the mediastinum, precise pathological diagnosis is required to determine the appropriate treatment strategy [2, 3]. Techniques to obtain tissue from the mediastinal tumours for pathological examination include approaches using needle biopsy, such as endoscopic ultrasound-guided fine needle aspiration and transbronchial needle biopsy. Furthermore, surgical techniques, such as anterior mediastinotomy, cervical mediastinoscopy, video assisted thoracoscopy and thoracotomy, can be utilized [3]. Ultrasound and computed tomography (CT) are the usual guiding methods of percutaneous needle biopsies, as these modalities enable accurate localization of the targeted lesion and visualization of the needle (4). In addition, image-guided percutaneous transthoracic needle biopsy enables access to practically all regions of the mediastinum [4]. CT is the preferred guiding method for mediastinal biopsy [5].

Because of essential vessels and organs in the mediastinum, biopsy may be challenging [6]. The mediastinum is limited by the thoracic inlet superiorly, the diaphragm inferiorly, the sternum anteriorly, the vertebral column posteriorly and the mediastinal pleura laterally [7]. According to the classical anatomical division system, the mediastinum is divided into a superior compartment and an inferior compartment by a plane extending from the fourth thoracic vertebra to the lower manubrium. The pericardium further subdivides the inferior compartment into an anterior, middle, and a posterior compartment [7], where the heart lies in the middle compartment.

Another classification system is the Felson classification which divides the mediastinum into an anterior, a middle and a posterior compartment. The lateral chest radiography is the basis for Felson's classification [8]. The posterior boundary of the anterior mediastinum is a line that runs from the thoracic inlet to the diaphragm in front of the trachea and behind the heart [8]. The posterior compartment is bordered anteriorly from the middle compartment by a line 1 cm behind the frontal borders of the vertebral column. The middle mediastinal compartment is localized between the anterior and posterior compartments [9].

Felson's anterior compartment is much larger than the anatomical anterior inferior compartment. The anatomical anterior inferior, middle inferior and part of the superior compartments are all included in Felson's anterior compartment [10].

CT is the most widely used method to examine mediastinal lesions [11]. Mediastinal lesions can be an incidental finding on CT [12]. The detailed image provided by a CT scan can localize even small lesions [13]. Furthermore, the CT scan enables the precise planning of the needle biopsy procedure, so that accidental puncture of vascular structures and vital organs can be avoided [13]. The most common biopsy complications are pneumothorax and pneumomediastinum. Other possible complications include haemoptysis, haemothorax, infections and air embolism [3]. Complication rates vary in the literature, and complication rates range between 3,8% and 17% [3, 9, 13–15]. The reported pneumothorax-rate varies between 1% and 10,3% [3, 9, 10, 15]. Diagnostic yield varies between 77% and 96% [3, 9, 10, 13–16]. Despite varying complication rates and diagnostic yield, there is agreement upon the importance of image guided percutaneous mediastinal core biopsy in the diagnosis of mediastinal lesions [3].

The scope of this study are the tumours referred for CT-guided percutaneous biopsy. Endobronchial ultrasound (EBUS) can be used to obtain access to subcarinal, paratracheal, and paraesophageal lymph nodes [4]. At Akershus University hospital, mediastinoscopies is a rarely performed procedure and it has mostly been replaced by (EBUS) and CT guided biopsies. Mediastinoscopy requires general anesthesia and not all tumours of the mediastinum are accessible by this procedure [4]. CT guided percutaneous biopsy only require local anaesthesia [7]. At Akershus University hospital, if a mediastinal tumour is inaccessible by EBUS, a CT

guided percutaneous biopsy will be performed. CT guided procedure is also preferred over EBUS when there is a need for larger biopsy size which often is necessary to make tumor marker analysis.

The purpose of this paper is to present our observations in regards to CT-guided percutaneous needle biopsy of mediastinal lesions with focus on complications and anatomical location, at a major university hospital in Norway.

## Materials and methods

### Sample and study design

We performed a retrospective analysis of 48 CT-guided percutaneous needle biopsies of mediastinal tumours, performed between April 2015 and August 2019, in total 53 months, at Akershus University Hospital.

To find the procedures to be included in the study, the local picture archiving and communication system was utilized to search for procedures containing the words "mediastinum" and "biopsy" in their procedure description. The resulting procedures were manually examined. Percutaneous biopsy procedures of lesions located in the mediastinum were included in the study.

The study was ethically approved by the regional ethical committee for medical and health research ethics (REK), with waived written consent because of the retrospective design of the study.

### CT-guided mediastinal biopsy procedure

Prior to the biopsy procedure, an antitussive and a mild sedative drug (5 mg dihydrocodeine and 5 mg diazepam orally), was administered. Most procedures were performed with Philips Brilliance 64 or Ingenuity 128 Core scanners, both with a tiltable gantry. While some procedures used Philips ICT 256 with a fixed gantry (Philips, Amsterdam, Netherlands). With the patient in the appropriate position in regards to the most safe and optimal trajectory path to the mediastinal lesion, a low-dose CT-scan was performed immediately prior to the biopsy procedure.

During the procedure a 17-G coaxial needle was inserted and usually positioned at the peripheral margin of the targeted lesion. This was done using local anaesthesia, intermittent CT fluoroscopy guidance and a stepwise technique. In some procedures, to access the lesion, the CT-gantry was tilted. Depending on the size of the lesion, respiratory movements and available scanner options, single CT slices with a thickness of 5, 7.5 or 10 mm were used. Usually, three or four needle insertions were performed. A reusable core (cutting) biopsy system with a 17G coaxial needle and an 18G biopsy needle (Bard®Magnum®, Bard Medical, Covington GA, USA) was used. The stroke length of the needle was set to either 15 or 22 mm, with the optimal stroke length determined for each individual case.

Immediately after the procedure, a full volume low-dose CT-scan was performed to detect any possible complications such as pneumothorax or lung haemorrhage. During the four hours of observation after the procedure, the patient was not allowed to drink or eat and had to lie flat. An upright, frontal projection chest x-ray was performed approximately two hours after the procedure. If a pneumothorax was symptomatic, if it evolved on the chest x-ray two hours after the procedure or if it occupied more than approximately 20–30% of the hemithorax, the radiologist performed a chest drainage using a pigtail catheter. In some cases, a thoracic surgeon inserted a Tru-Close Thoracic Vent ® (UreSil, Skokie IL, USA) or a chest tube.

## Target characteristics

The individual lesions were examined on a contrast-enhanced prebiopsy CT-scan. Their localisation within the mediastinum, both according to the anatomical division and Felson's division, was determined by a thoracic radiologist with more than 10 years experience. If the lesion was located in more than one compartment it was localised according to the compartment were the majority of the lesion was located. Each lesions morphology was decided. The size of the lesion was determined in the axial plane by measuring the two largest diameters of the lesion perpendicular to each other. In the craniocaudal direction, the distance between the superior and inferior boarder of the (continuous) lesion was registered. The three resulting measures were multiplied and divided by two to estimate the lesions volume.

The obtained tissue was sent to our hospitals Department for Pathology for histological examination. The final diagnosis was retrieved from the pathologists written reports. The histology of the biopsies was subdivided into following categories: benign, carcinoma, thymoma, lymphoma and other malignancies.

## Registration of complications

The biopsy procedure description and the post-biopsy CT-scan and x-ray were manually examined to register complications. Pneumothorax, pneumomediastinum, chest wall emphysema and haemothorax on CT and/or chest X-ray were registered. In case of pneumothorax, it was noted if chest drainage was performed. Complications like parenchymal lung haemorrhage and mediastinal haemorrhage were registered from the post procedure CTs. Even the smallest observable signs of air-leak or haemorrhage on CT was registered as a complication.

## Statistical analysis

The results were analysed using SPSS statistical software version 26 with a level of significance set to 5%. Associations between nominal variables such as mediastinal compartments and pathology groups were analysed using chi square Cramér's V–test.

# Results

48 biopsy-procedures were performed. There were 45 patients, 29 men and 16 women, with an age range from 25 years to 92 years, a median age of 58 years and an average age of 60,5 years. For the men the age ranged between 25 years and 83 years; with the median age of 58 years. The women had a median age of 60,5 years and an age range from 35 years to 92 years.

According to the traditional anatomical mediastinal division, 11 lesions were found in the superior mediastinum, 30 of the lesions were in the inferior anterior mediastinum, and only 4 lesions were in the inferior posterior mediastinal region. No lesions were found in the inferior middle mediastinum. However, when sorting the lesion by Felson's mediastinal division criteria, 40 lesions were situated in the anterior compartment, 4 lesions in the middle compartment and 1 lesion was localized in the posterior compartment (Table 1).

The estimated volume of the lesions ranged from 1.4 $cm^3$ to 2195 $cm^3$ with a median volume of 26.5 cm3 and an average volume of 154 $cm^3$.

On CT the morphology of the lesions was homogeneous in 40% (18/45) of the cases and heterogeneous in the remaining 60% (27/45). The border was smooth in 42% (19/45) and irregular in 58% (26/45) of the lesions. Calcification was seen on CT in 13% (6/45) of the lesions. These features along with volume of the tumour could not distinguish whether the tumours were malignant or benign in multivariate analysis all with p values above 0.132.

**Table 1. Characteristics of patients (n = 45), lesions (n = 45) and procedures (n = 48).**

|  | n |  |
|---|---|---|
| Number (%) unless otherwise specified |  |  |
| Patients |  |  |
| Age (years), mean (range) | 60,5 | (25–92) |
| Sex |  |  |
| Male | 29 | (64,4%) |
| Female | 16 | (35,6%) |
| Lesion characteristics (45) |  |  |
| Lesion volume (cm^3), mean (range) | 153.9 | (1.4–2195) |
| Morphology |  |  |
| Heterogenous | 27 | (60%) |
| Homogenous | 18 | (40%) |
| Calcification | 6 | (13,3%) |
| Smooth border | 19 | (42,2%) |
| Irregular border | 26 | (57,8%) |
| Mediastinal compartments, anatomical (45) |  |  |
| Superior | 11 | (24,4%) |
| Inferior anterior | 30 | (66,6%) |
| Inferior middle | 0 | (0%) |
| Inferior posterior | 4 | (9%) |
| Mediastinal compartments, Felson (45) |  |  |
| Anterior | 40 | (88,9%) |
| Middle | 4 | (8,9%) |
| Posterior | 1 | (2,2%) |
| Procedure characteristics (48) |  |  |
| Access * |  |  |
| Parasternal | 29 | (60,4%) |
| Paravertebral | 3 | (6,3%) |
| Transsternal | 2 | (4,2%) |
| Transpulmonal | 20 | (41,7%) |
| Complications |  |  |
| Pneumothorax | 12 | (25%) |
| Pneumomediastinum | 18 | (37,5%) |
| Chest wall emphysema | 2 | (4,2%) |
| Lung haemorrhage | 6 | (12,5%) |
| Mediastinal haemorrhage | 9 | (18,8%) |
| Hemothorax | 1 | (2,1%) |
| **Chest drainage** | **4** | **(8,3%)** |

*Five procedures were both parasternal and transpulmonal. One procedure was both paravertebral and transpulmonal.

The parasternal approach was used in most of the biopsy procedures (29/48) (Fig 1). Para-vertebral approach was chosen in three procedures and transsternal approach was used in two procedures. A transpulmonary approach, which involves penetration of the lung and visceral pleura by the needle, was performed in 20 of the procedures. Five procedures were both para-sternal and transpulmonal, and one procedure was both paravertebral and transpulmonal.

Pneumothorax only occurred when transpulmonal access was performed 20 transpulmonal procedures were performed and 12 of them resulted in pneumothorax, Thus, pneumothorax

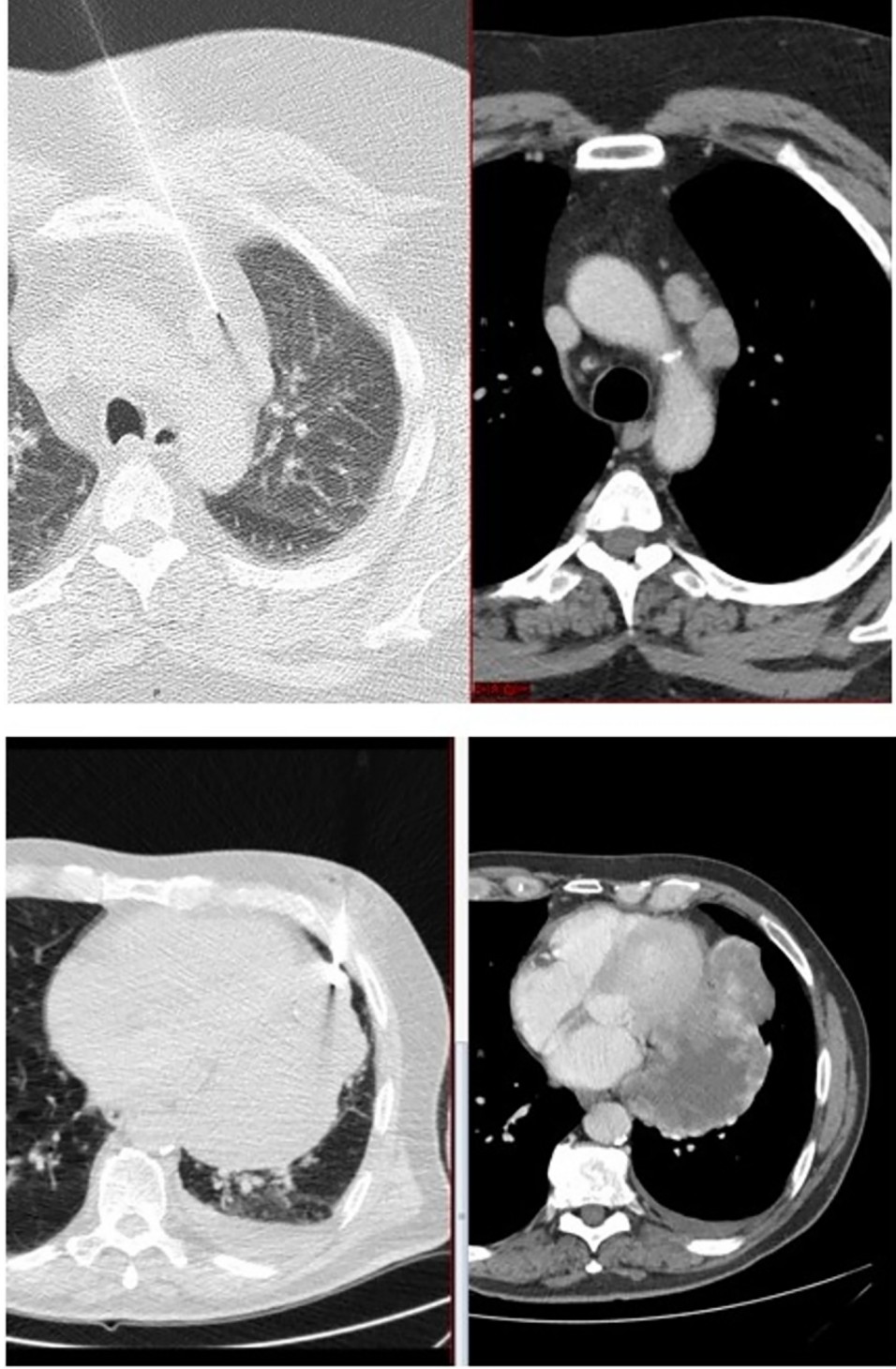

**Fig 1. Two images of biopsies.** Top: Shows a biopsy procedure with parasternal access. The lesion is located in Felson's anterior compartment and in the superior compartment according to the anatomical division system. The diagnosis was metastasis from prostate cancer. Bottom: Shows a biopsy procedure with transpulmonal access. The lesion is located in Felson's anterior compartment and in the inferior anterior compartment according to the anatomical division system. The diagnosis was sarcoma.

**Table 2. Presents number and type complications within each compartment.**

| Compartment | Procedure | Complication | | | | | Types of complications | | | |
|---|---|---|---|---|---|---|---|---|---|---|
| | | No | Yes | | Pneumo-thorax | Pneumo-mediastinum | Chest wall emphysema | Lung parenchymal bleeding | Mediastinal haemorrhage | Haemo-thorax |
| | (N) | (n) | (n) | (n) | | | | | | |
| **Inferior anterior** | **32** | **7** | **33** | | **10** | **14** | **1** | **2** | **6** | **0** |
| Singular complication | | | | 17 | 9 | 6 | | 1 | 1 | |
| Several complications | | | | | | | | | | |
| Event a | | | | 5 | | 5 | | | 5 | |
| Event b | | | | 1 | 1 | 1 | | | | |
| Event c | | | | 1 | | 1 | 1 | | | |
| Event d | | | | 1 | | 1 | | 1 | | |
| **Inferior posterior** | **4** | **2** | **4** | | **1** | **0** | **0** | **2** | **0** | **1** |
| Singular complication | | | | 1 | | | | 1 | | |
| Several complications | | | | 1 | 1 | | | 1 | | 1 |
| **Superior** | **12** | **4** | **11** | | **1** | **4** | **1** | **2** | **3** | **0** |
| Singular complication | | | | 5 | | 3 | | | 2 | |
| Several complications | | | | | | | | | | |
| Event a | | | | 1 | 1 | | | 1 | | |
| Event b | | | | 1 | | 1 | | | 1 | |
| Event c | | | | 1 | | | 1 | 1 | | |
| Σ: | **48** | **13** | **48** | | **12** | **18** | **2** | **6** | **9** | **1** |

occurred in 60% (12/20) of the transpulmonal procedures. Nine of the pneumothoraxes were still visible on x-ray two hours after the procedure. Four of the pneumothoraxes required chest drainage, this accounts for 20% of the transpulmonary procedures (4/20). Pneumomediastinum was detected with 18 of the procedures (37,5% (18/48)), but did not result in any additional complications (Table 2).

Chest wall emphysema was seen after two procedures. Six procedures led to lung haemorrhage. Mediastinal haemorrhage was observed after nine procedures and haemothorax after one procedure (Table 2). These complications were minor and did not require further treatment or follow-up. There were no fatalities.

Adequate tissue material, i.e. enough material for the pathologists to make a diagnostic conclusion, was obtained in 46 out of 48 procedures. The remaining two biopsies were inconclusive. As shown in Table 3, the histology revealed that 80% (36/45) of the lesions were malignant and 20% (9/45) were benign. Thymomas, carcinomas (primary and metastasis) and lymphomas accounted for most of the malignant lesions.

Four patients underwent a repeated biopsy procedure. Two of these patients had a diagnostic first biopsy, but since the pathological diagnosis (normal thymus tissue) and the anticipated diagnosis did not correspond, a second biopsy was conducted to confirm that the lesions were benign. Two procedures were initially inconclusive and had to be re-biopsied. One of these re-biopsy procedures is included in this study, and the diagnosis turned out to be thymoma. The other re-biopsy procedure however is not included in this study because it was out of the

**Table 3. Histological analysis of the lesions.**

|  | n | Percentage |
|---|---|---|
| Thymoma | 14 | 29,2% |
| Lymphoma | 7 | 14,6% |
| Carcinoma (primary or metastasis) | 11 | 22,9% |
| Other malignancies | 3 | 6,3% |
| Benign | 11 | 22,9% |
| Inconclusive | 2 | 4,2% |
| **Total** | **48** | **100%** |

timeframe of the study. The diagnosis of which was lymphoma. Hence, the study includes 45 patients with a total of 48 procedures.

Tables 4 and 5 show the correlation between the tissue diagnosis of the 45 lesions and their localisation, according to the anatomical division and Felson's division respectively. The re-biopsied lesions are registered only once.

Table 4 shows the tissue diagnosis in relation to anatomical compartments. All thymomas are in the anatomical inferior anterior compartment and account for 47% of the lesions in this compartment. The anatomical division shows a varying distribution of benign lesions, with 18,2% of the lesions in the superior compartment, 17% in the anterior compartment and 50% in the inferior posterior compartment. Table 5 shows the relation between tissue diagnosis and Felson's compartments. With this division 89% of the lesions, both benign and malignant, lie in the anterior compartment.

Cramér's V -test shows a significant ($p = 0,006$), moderate to high (Cramér's V = 0,49) association between anatomical compartments and tissue diagnosis. The data does not show significant association between Felson's compartments and tissue diagnosis (Cramér's V = 0,36, $p = 0,166$).

**Table 4. Tissue diagnosis and anatomical mediastinal compartments.**

|  |  |  | Anatomical mediastinal classification | | | |
|---|---|---|---|---|---|---|
|  |  |  | Inferior anterior | Inferior posterior | Superior | Total |
| **Tissue diagnosis** | Thymoma | Count | 14 | 0 | 0 | 14 |
|  |  | %within compartment | 46,7% | 0,0% | 0,0% | 31,1% |
|  |  | % within total | 31,1% | 0,0% | 0,0% | 31,1% |
|  | Lymphoma | Count | 5 | 2 | 1 | 8 |
|  |  | %within compartment | 16,7% | 50,0% | 9,1% | 17,8% |
|  |  | % within total | 11,1% | 4,4% | 2,2% | 17,8% |
|  | Carcinoma (primary and metastasis) | Count | 4 | 0 | 7 | 11 |
|  |  | %within compartment | 13,3% | 0,0% | 63,6% | 24,4% |
|  |  | % within total | 8,9% | 0,0% | 15,6% | 24,4% |
|  | Other malignancy | Count | 2 | 0 | 1 | 3 |
|  |  | %within compartment | 6,7% | 0,0% | 9,1% | 6,7% |
|  |  | % within total | 4,4% | 0,0% | 2,2% | 6,7% |
|  | Benign | Count | 5 | 2 | 2 | 9 |
|  |  | %within compartment | 16,7% | 50,0% | 18,2% | 20,0% |
|  |  | % within total | 11,1% | 4,4% | 4,4% | 20,0% |
| **Total** |  | Count | 30 | 4 | 11 | 45 |
|  |  | %within compartment | 100,0% | 100,0% | 100,0% | 100,0% |
|  |  | % within total | 66,7% | 8,9% | 24,4% | 100,0% |

**Table 5. Tissue diagnosis and Felson's mediastinal compartments.**

| | | | Anterior | Middle | Posterior | Total |
|---|---|---|---|---|---|---|
| | | | | | Felson mediastinal compartmets | |
| **Tissue diagnosis** | Thymoma | Count | 14 | 0 | 0 | 14 |
| | | %within compartment | 35,0% | 0,0% | 0,0% | 31,1% |
| | | % within total | 31,1% | 0,0% | 0,0% | 31,1% |
| | Lymphoma | Count | 6 | 2 | 0 | 8 |
| | | %within compartment | 15,0% | 50,0% | 0,0% | 17,8% |
| | | % within total | 13,3% | 4,4% | 0,0% | 17,8% |
| | Carcinoma (primary and metastasis) | Count | 11 | 0 | 0 | 11 |
| | | %within compartment | 27,5% | 0,0% | 0,0% | 24,4% |
| | | % within total | 24,4% | 0,0% | 0,0% | 24,4% |
| | Other malignancy | Count | 3 | 0 | 0 | 3 |
| | | %within compartment | 7,5% | 0,0% | 0,0% | 6,7% |
| | | % within total | 6,7% | 0,0% | 0,0% | 6,7% |
| | Benign | Count | 6 | 2 | 1 | 9 |
| | | %within compartment | 15,0% | 50,0% | 100,0% | 20,0% |
| | | % within total | 13,3% | 4,4% | 2,2% | 20,0% |
| **Total** | | Count | 40 | 4 | 1 | 45 |
| | | %within compartment | 100,0% | 100,0% | 100,0% | 100,0% |
| | | % within total | 88,9% | 8,9% | 2,2% | 100,0% |

## Discussion

Adequate tissue material was obtained in 46 procedures, which gives a diagnostic yield of 96%. The diagnostic yield is comparable with observations in other studies [3, 13, 14, 16]. Hence, percutaneous CT-guided mediastinal biopsy is an effective procedure.

This study has a relatively high complication rate compared to other studies [3, 9, 13–15]. This is most likely explained by our definition of complications, as we defined even the smallest observable signs of air-leak or haemorrhage on CT as a complication.

As CT is a highly sensitive imaging technique, a lot of non-significant complications have been registered even though they had no clinical relevance.

Pneumothorax was observed in 60% of the transpulmonary procedures, one fourth of the total procedures. We report a higher pneumothorax -rate than the rates in similar studies [3, 9, 10, 15]. This is again likely explained by our definition of complications. Relatively few patients needed chest drainage. In accordance with previously published material (10), pneumothorax only occurred when transpulmonal approach was utilized. As many as 60% of the transpulmonal approaches resulted in pneumothorax. Most of the pneumothoraxes were self-resolving, with chest drainage performed in only 4 out of 12 cases, 33% of the pneumothoraxes and approx. 8% of the total number of procedures. Despite the substantial risk of pneumothorax when transpulmonary access is chosen, the percutaneous CT-guided mediastinal biopsy is considered a safe procedure. Other complications were minor and demanded no further follow up or treatment. None of the procedures had a fatal outcome. Based on the dataset, it can therefore be concluded that percutaneous CT-guided mediastinal biopsy is a safe procedure.

The scope of this study is to analyse the tumours referred for CT-guided percutaneous biopsy. As mentioned, at Akershus University hospital tumours inaccessible by EBUS or tumours accessible by EBUS, but where there is need for a larger tissue sample, are referred for CT-guided percutaneous biopsy.

Our data suggests that when a tumour has been selected for CT-guided percutaneous needle biopsy, the likelihood of a malignant diagnosis is higher, the further anteriorly the lesion is located within the mediastinum. In the anterior inferior compartment, as many as 83% of the lesions were malignant. All the 14 thymomas in our study were found in the anterior inferior compartment, where they commonly occur [7] and accounted for 56% of the malignant lesions in this compartment. There were 82% malignant lesions in the superior compartment, of which carcinomas accounted for 64%. Half of the relatively few lesions in the posterior compartment were malignant.

There was no significant association between tissue diagnosis and Felson's division system. Almost all the lesions were located in the Felson anterior compartment. The anatomical anterior inferior, middle inferior and part of the superior compartments are all included in Felson's anterior compartment. Hence, Felson's mediastinal compartments does not have any applicable value in regard to predicting whether a mediastinal tumors selected for CT-guided needle biopsy is benign or malignant We therefore recommend the anatomical compartment system for assessment and characterization of mediastinal tumours.

For lesions accessed by CT-guided needle biopsy, the data shows a significant, moderate to high association between tissue diagnosis and the classical anatomical division. The data suggests that for mediastinal tumours selected for CT guided biopsy, the lesions anatomical compartment can indicate the probability of a benign or malignant diagnosis. Whether the association between the anatomical localisation and the likelihood of a malignant diagnosis also is applicable for mediastinal tumours not selected for CT-guided percutaneous biopsy, is uncertain and it requires further research to elaborate this hypothesis. CT morphology features along with tumour volume could not distinguish between malignant and benign tumours in multivariate statistical analysis. Hence, «tissue is the issue».

A limitation of this study is the relatively small number of biopsy procedures and the retrospective design.

## Conclusion

Our study found that CT guided mediastinal biopsy is a safe procedure with a high diagnostic yield of 96%. We therefore recommend CT guided biopsy for diagnostic workup of mediastinal tumours. Pneumothorax (60% of the transpulmonary procedures), pneumomediastinum (38%) and chest drainage (8%) were the main complications. We had no fatal outcome from the mediastinal biopsies.

We recommend use of anatomical compartment division of mediastinum compared to Felson's division—we found significant association between anatomical compartment and tissue diagnosis for tumours selected for CT-guided percutaneous needle biopsy. The likelihood of a malignant diagnosis is higher when the lesion is located in the anterior part of the mediastinum. Using anatomical mediastinal compartments most carcinomas were in the superior compartment, whereas all thymomas were found in the inferior anterior compartment.

## Acknowledgments

Language revision was performed by MD Kristina Shanley, Norway.

## Author Contributions

**Conceptualization:** Ingegjerd Kristina Skretting, Espen Asak Ruud, Haseem Ashraf.

**Data curation:** Ingegjerd Kristina Skretting, Espen Asak Ruud, Haseem Ashraf.

**Formal analysis:** Ingegjerd Kristina Skretting, Espen Asak Ruud, Haseem Ashraf.

**Investigation:** Ingegjerd Kristina Skretting, Espen Asak Ruud, Haseem Ashraf.

**Methodology:** Espen Asak Ruud, Haseem Ashraf.

**Project administration:** Ingegjerd Kristina Skretting, Haseem Ashraf.

**Supervision:** Espen Asak Ruud, Haseem Ashraf.

**Validation:** Espen Asak Ruud, Haseem Ashraf.

**Visualization:** Haseem Ashraf.

**Writing – original draft:** Ingegjerd Kristina Skretting.

**Writing – review & editing:** Ingegjerd Kristina Skretting, Espen Asak Ruud, Haseem Ashraf.

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
