## [Decision Letter · Decision Letter 0]

19 May 2022

PONE-D-21-29914

Diagnostic yield, complications, pathology and anatomical features in CT-guided percutaneous needle biopsy of mediastinal tumours.

PLOS ONE

Dear Dr. Ashraf,

Thank you for submitting your manuscript to PLOS ONE. After careful consideration, we feel that it has merit but does not fully meet PLOS ONE’s publication criteria as it currently stands. Therefore, we invite you to submit a revised version of the manuscript that addresses the points raised during the review process.

We look forward to receiving your revised manuscript.

Kind regards,

Luka Brcic

Academic Editor

PLOS ONE

**Journal requirements:**

“NO”

3. Thank you for stating the following in your manuscript:

“The study was funded by internal research funding from the hospital.”

“No”

“NO authors have competing interests”

5. PLOS requires an ORCID iD for the corresponding author in Editorial Manager on papers submitted after December 6th, 2016. Please ensure that you have an ORCID iD and that it is validated in Editorial Manager. To do this, go to ‘Update my Information’ (in the upper left-hand corner of the main menu), and click on the Fetch/Validate link next to the ORCID field. This will take you to the ORCID site and allow you to create a new iD or authenticate a pre-existing iD in Editorial Manager. Please see the following video for instructions on linking an ORCID iD to your Editorial Manager account: https://www.youtube.com/watch?v=_xcclfuvtxQ.

6. Please amend your authorship list in your manuscript file to include list of authors.

7. We note that you have included the phrase “data not shown” in your manuscript. Unfortunately, this does not meet our data sharing requirements. PLOS does not permit references to inaccessible data. We require that authors provide all relevant data within the paper, Supporting Information files, or in an acceptable, public repository. Please add a citation to support this phrase or upload the data that corresponds with these findings to a stable repository (such as Figshare or Dryad) and provide and URLs, DOIs, or accession numbers that may be used to access these data. Or, if the data are not a core part of the research being presented in your study, we ask that you remove the phrase that refers to these data.

Reviewers' comments:

Reviewer's Responses to Questions

**Comments to the Author**

1. Is the manuscript technically sound, and do the data support the conclusions?

Reviewer #1: No

Reviewer #2: Yes

2. Has the statistical analysis been performed appropriately and rigorously? 

Reviewer #1: No

Reviewer #2: Yes

3. Have the authors made all data underlying the findings in their manuscript fully available?

Reviewer #1: Yes

Reviewer #2: Yes

4. Is the manuscript presented in an intelligible fashion and written in standard English?

Reviewer #1: Yes

Reviewer #2: Yes

5. Review Comments to the Author

Reviewer #1: Thank you for the opportunity to review the article entitled “Diagnostic yield, complications, pathology and anatomical features in CT-guided percutaneous needle biopsy of mediastinal tumors”. The authors retrospectively describe their experience using CT-guided needle biopsy to sample mediastinal tumors. While the data is interesting, there are major limitations to this study that needs to be clarified by the authors, some of which are inherent to the nature of the study. While the authors did mention the small number and retrospective nature of their study as a limitation further elaboration is required.

The authors rightfully describe other ways of sampling mediastinal tumors, such as endoscopic and surgical techniques. The authors should describe whether these techniques are used at their institution. As this is a retrospective study, it is important to have a sense of whether the percutaneous approach is used for all patients with mediastinal tumor or a carefully selected subset.

Along the same lines, most of the lesions sampled were in the inferior anterior mediastinum (30 lesions, %63 of procedures), with no cases in the inferior middle mediastinum. It would be helpful for the reader to know what the risk of complications with each compartment as the concern would be that the data may only reflect the safety of sampling lesions in the inferior anterior mediastinum rather than the whole mediastinum.

The authors should clearly state the risk of pneumothorax. The overall risk of pneumothorax was (25%, 12 cases) with intervention required in 4 cases (8%). Arguably, the risk of pneumothorax and the need for intervention should be calculated purely for the transpulmonary approach (60% of cases had pneumothorax, 20% required intervention)

Selection bias may explain why there were no mediastinal lesions sampled in the inferior middle mediastinum. Assuming that the institution has an endoscopy/bronchoscopy service, the lesions in the inferior middle mediastinum may have been sampled endoscopically given the proximity to major vascular structures. This again affects the way the results should be interpreted. Similarly, the recommendation to use compartment system vs the Felson system as result may not necessarily be accurate.

I do not believe that the statement that the likelihood of malignant diagnosis is higher the further anteriorly the lesion is located in the mediastinum is valid unless all the mediastinal lesions at the institution are sampled via the percutaneous approach. Otherwise the conclusion is biased toward the lesions that were easier to access via the percutaneous approach compared to other approaches (endoscopic/Surgical)

The authors mention that two procedures were initially inconclusive and had to be re-biopsied. However, only one re-biopsy was included in the study. Can the authors explain why the other re-biopsy was not included (out of the study timeframe vs another sampling approach was used)?

Minor comment

Introduction, 5th paragraph. As the authors are referencing more than 3 studies, would suggest changing the sentence to state that the complication rates range between 3-17%.

Reviewer #2: Dear Authors,

Thank You for an interesting article. The theme of this article is radiological, yet in the article, I do not find an explanation why is the CT imaging method of choice for percutaneous needle biopsy of this region and what are the advantages of this method in comparison with other imaging methods. In Figure 1. You show the needle tip in the lung window and the pathology in a mediastinal window. Is there any practical explanation for it (considering that the size of the lesion depending on the window can be underrated or overrated, the target components of the lesion during the biopsy are poorly discriminated in the lung window)? How do You define the specific low-dose CT examination You used in the study? Postinterventional control radiography was performed. Is the sensitivity of this method in comparison to CT a limitation for the detection of postinterventional complications? In which cases were chest drainage and surgical drainage performed? Regarding the diagnostic yield and adequate tissue material, did You find any statistical difference related to the tissue structure of the lesion (solid vs. cystic-solid)? How was adequate tissue material defined?

In the discussion, much information pertains to results. Therefore, I would suggest rewriting the discussion and comparing more in detail the study results with current literature.

The text also needs some language adaptation and consistency in writing numbers (for instance - 2 or two).

6. PLOS authors have the option to publish the peer review history of their article (what does this mean?). If published, this will include your full peer review and any attached files.

Reviewer #1: **Yes: **Ala Eddin Sagar

Reviewer #2: No

---

## [Author Response · Author response to Decision Letter 0]

16 Oct 2022

Point by point response has been attached.

---

## [Editor Report · Decision Letter 1]

24 Oct 2022

Diagnostic yield, complications, pathology and anatomical features in CT-guided percutaneous needle biopsy of mediastinal tumours.

PONE-D-21-29914R1

Dear Dr. Ashraf,

We’re pleased to inform you that your manuscript has been judged scientifically suitable for publication and will be formally accepted for publication once it meets all outstanding technical requirements.

Kind regards,

Luka Brcic

Academic Editor

PLOS ONE
---

## [Editor Report · Acceptance letter]

26 Oct 2022

PONE-D-21-29914R1 

Diagnostic yield, complications, pathology and anatomical features in CT-guided percutaneous needle biopsy of mediastinal tumours. 

Dear Dr. Ashraf:

I'm pleased to inform you that your manuscript has been deemed suitable for publication in PLOS ONE. Congratulations! Your manuscript is now with our production department. 

Kind regards, 

on behalf of

Dr. Luka Brcic 

Academic Editor

PLOS ONE